# Patient-Derived Organoids of Cholangiocarcinoma

**DOI:** 10.3390/ijms22168675

**Published:** 2021-08-12

**Authors:** Christopher Fabian Maier, Lei Zhu, Lahiri Kanth Nanduri, Daniel Kühn, Susan Kochall, May-Linn Thepkaysone, Doreen William, Konrad Grützmann, Barbara Klink, Johannes Betge, Jürgen Weitz, Nuh N. Rahbari, Christoph Reißfelder, Sebastian Schölch

**Affiliations:** 1Junior Clinical Cooperation Unit Translational Surgical Oncology (A430), German Cancer Research Center (DKFZ), 69120 Heidelberg, Germany; christopher.maier@dkfz.de (C.F.M.); lei.zhu@dkfz.de (L.Z.); 2Department of Surgery, Medical Faculty Mannheim, Universitätsmedizin Mannheim, Heidelberg University, 68167 Mannheim, Germany; nuh.rahbari@umm.de (N.N.R.); christoph.reissfelder@umm.de (C.R.); 3Department of Gastrointestinal, Thoracic and Vascular Surgery, Medizinische Fakultät Carl Gustav Carus, Technische Universität Dresden, 01307 Dresden, Germany; lahirin@flowcell.co (L.K.N.); daniel.kuehn@elblandkliniken.de (D.K.); susan.kochall@ukdd.de (S.K.); may-linn.thepkaysone@ukdd.de (M.-L.T.); juergen.weitz@ukdd.de (J.W.); 4Core Unit for Molecular Tumor Diagnostics (CMTD), National Center for Tumor Diseases (NCT) Partner Site Dresden, 01307 Dresden, Germany; doreen.william@nct-dresden.de (D.W.); Konrad.Gruetzmann@uniklinikum-dresden.de (K.G.); Barbara.Klink@lns.etat.lu (B.K.); 5German Cancer Consortium (DKTK) and German Cancer Research Center (DKFZ), 69120 Heidelberg, Germany; 6National Center of Genetics, Laboratoire National de Santé (LNS), 3555 Dudelange, Luxembourg; 7Junior Clinical Cooperation Unit Translational Gastrointestinal Oncology and Preclinical Models (B440), German Cancer Research Center (DKFZ), 69120 Heidelberg, Germany; j.betge@dkfz.de; 8Department of Medicine II, Medical Faculty Mannheim, Universitätsmedizin Mannheim, Heidelberg University, 68167 Mannheim, Germany

**Keywords:** cholangiocarcinoma, translational surgical oncology, organoids, patient-derived organoids, xenograft model, orthotopic xenograft, response prediction, next-generation sequencing, precision medicine

## Abstract

Cholangiocarcinoma (CC) is an aggressive malignancy with an inferior prognosis due to limited systemic treatment options. As preclinical models such as CC cell lines are extremely rare, this manuscript reports a protocol of cholangiocarcinoma patient-derived organoid culture as well as a protocol for the transition of 3D organoid lines to 2D cell lines. Tissue samples of non-cancer bile duct and cholangiocarcinoma were obtained during surgical resection. Organoid lines were generated following a standardized protocol. 2D cell lines were generated from established organoid lines following a novel protocol. Subcutaneous and orthotopic patient-derived xenografts were generated from CC organoid lines, histologically examined, and treated using standard CC protocols. Therapeutic responses of organoids and 2D cell lines were examined using standard CC agents. Next-generation exome and RNA sequencing was performed on primary tumors and CC organoid lines. Patient-derived organoids closely recapitulated the original features of the primary tumors on multiple levels. Treatment experiments demonstrated that patient-derived organoids of cholangiocarcinoma and organoid-derived xenografts can be used for the evaluation of novel treatments and may therefore be used in personalized oncology approaches. In summary, this study establishes cholangiocarcinoma organoids and organoid-derived cell lines, thus expanding translational research resources of cholangiocarcinoma.

## 1. Introduction

Cholangiocarcinoma (CC) presents a heterogeneous group of malignancies originating from the biliary epithelium accounting for about 3% of all gastrointestinal cancers. According to its anatomical location, CC can be classified into distal (dCC), perihilar (pCC) and intrahepatic (iCC) tumors [1,2]. Although iCC is the second most common primary liver cancer after hepatocellular carcinoma (HCC), it is the rarest subtype as most CC arises in the perihilar region (50–67% pCC, 27–42% dCC, 6–8% iCC) [3,4]. Biliary tract diseases leading to chronic inflammation such as primary sclerosing cholangitis, bile duct cysts, hepatolithiasis, or opisthorchiasis are known to increase the risk of CC [5,6]. Besides, chronic biliary inflammation and chronic cholestasis, together or separately, appear to be critical factors in cholangiocarcinogenesis [7]. The prognosis of CC remains dismal, with surgery being the only potentially curative treatment. In advanced CC, the availability of effective systemic treatment options is still limited [8]. As a result, five-year overall survival is low, ranging from 2% (metastatic CC of any location) to 25 % (resectable iCC) [9].

Numerous somatic alterations were described in CC affecting oncogenes (e.g., *KRAS*), tumor suppressor genes (*TP53*, *SMAD4*), epigenetic factors (*IDH1/2*), chromatin-modifying genes (e.g., *ARID1A*, *BAP1*, *ERBB2*, *FGF2*, and *PBMR1*) [10,11,12,13,14]. The abnormal expression of *KRAS* and *TP53* results in a more aggressive CC phenotype [15]. Furthermore, the mutational profiles of CCs differ greatly depending on etiology, ethnicity, and anatomical location [11,12,16]. For example, *KRAS* mutations are more common in pCCs (pCC 22–53%, iCC 9–17%), while *IDH1/2* mutations are more frequently present in iCCs [10,17]. The numerous somatic mutations present in CC, including IDH1 mutations, FGFR2 or NTRK fusions, HER2 amplifications, and others, make it a particularly target-rich tumor that may be exploited by specific inhibitors [13,18]. Testing these novel strategies and analyzing response mechanisms in preclinical models of CC to select the best strategies for subsequent clinical trials represents an important research topic.

Preclinical cancer models, such as classical 2D cell lines (henceforth referred to as CCLs) and patient-derived xenografts (PDXs), are essential tools for cancer research and therefore aid in developing potential treatment options [19,20,21]. However, CC-CCLs are notoriously hard to establish; as a result, the majority of preclinical research on CC has been performed in only two cell lines, EGI-1 and TFK-1 (both established from extrahepatic CC). It is conceivable that these two cell lines may not sufficiently reflect the molecular biology of CC [22,23,24,25].

Organoid culture is an emerging three-dimensional (3D) cell culture technique that allows for more self-organization of tumor cells than CCLs, thus better mimicking the complexity of the parental disease [26]. Human cancer organoids largely maintain the histological features, expression profile, and genomic landscape of their corresponding primary tissues, making them suitable for translational studies and identifying treatable molecular alterations in the context of personalized medicine [27,28]. Depending on the tumor entity, patient-derived organoids (PDOs) can be established from resected tumor tissues or core needle biopsies with higher success rates than classical CCLs, making PDOs an attractive in vitro model, especially for CC research [27,29,30]. Only a few PDO protocols have been published for cholangiocarcinoma with varying success rates, demonstrating the need for further refinement of this culture method for CC [27,31,32,33]. Here, we present a series of newly established human cholangiocarcinoma PDOs, report culture protocols, the genomic landscape of CC models, their applicability for xenograft experiments, and their response to drug treatment with current chemotherapeutic agents. In addition, we describe the successful transfer of established organoids into table 2D cultures, allowing easier handling and high-throughput experiments using organoid-derived 2D cell lines. In summary, this research demonstrates the stable culturing of cholangiocarcinoma and its translational research potentials.

## 2. Results

### 2.1. Establishment of Human Cholangiocarcinoma Organoid Lines from Surgical Specimens

We aimed to establish a standardized protocol for obtaining organoid cultures from CC surgical resection specimens. After mechanical and enzymatic digestion of the cholangiocarcinoma tumor tissue, suspensions were filtered, differential centrifuged, and seeded with Matrigel (Figure 1A). There was no robust cholangiocarcinoma organoid culture medium published when this project started (January 2015), so our organoids were initially cultured in an established human gastric cancer organoid medium [34], in which the first batch of organoids (*n* = 9) could not be maintained for more than two passages. Based on a trial-and-error method following in-depth literature research, the culture medium was optimized (by adding a Rho-associated protein kinase (ROCK) inhibitor, forskolin, insulin, transferrin, and selenite), leading to the generation of stable CC organoid cultures that could be maintained long-term. Two lines (P68 and P83) were selected for more in-depth characterization in this study (Table 1).

The primary tumor of P68 was a metastatic iCC, P83 was derived from a perihilar tumor, and a stable third CC organoid line was established using TFK-1, a commercially available human dCC-CCL [35] (Table 2). Using the same protocol, we were also able to generate organoids from human non-malignant bile duct mucosa (P119), which could be maintained for a few passages and were thus used as controls.

Morphologically, P68 organoids exhibited a small hollow cystic growth pattern with an outer epithelium up to 20 µm of thickness and an average diameter of 150–200 µm next to organoids with grape-like architecture (Figure 1B). P83 organoids grew as large hollow cystic structures with an average diameter of 250–300 µm and a thick outer lining up to 25 µm with some protrusions (Figure 1C). The control organoids of TFK-1 developed an average diameter of 200–400 µm and an outer epithelial lining of 15–25 µm without budding (Figure 1D).

#### Organoid-Derived 2D Cell Lines

We next tested whether it is possible to transfer established stable PDOs to 2D cell culture. Initially, PDOs were cultured in an organoid medium without Matrigel. Cells readily attached to the surface and grew as 2D cell lines. It was possible to maintain stable CCLs after gradually converting the organoid medium to a CCL medium. A mixture of 2/3 Dulbecco’s Modified Eagle’s Medium (DMEM)/20% fetal calf serum (FCS) and 1/3 Keratinocyte serum-free medium (K-SFM), and supplemented with 100 U/mL penicillin/streptomycin, was finally used as a culture medium, allowing the permanent 2D culture of the CC PDOs [36] (Appendix A).

The viability, proliferation, and metabolic activity of PDO-derived CCLs were measured using PrestoBlue cell viability reagent (Invitrogen) (Appendix A). While TFK-1 exhibited the fastest proliferation with an estimated doubling time of 38.5 ± 2.0 h in accordance with the literature [32], P68 and P83 cell lines presented slower growth in 2D with doubling times of 51.1 ± 5.0 h and 81.7 ± 12.7 h, respectively. All cell lines grew as adherent, epithelial monolayers in typical cobblestone-like patterns. The P68 cell line showed variably sized polygonal cells with most cells adherent and a small fraction of floating vital cells (Figure 1E). P83 grew as small and tightly packed, round or polygonal cells next to large, polygonal cells (Figure 1F). The control TFK-1 cell line showed uniformly sized polygonal cells as described before (Figure 1G) [32].

### 2.2. Patient-Derived Organoid-Based Xenografts of Cholangiocarcinoma

Next, we evaluated the tumorigenicity of the PDOs in a murine xenograft experiment. Organoids were injected subcutaneously (bilateral thighs) or orthotopically (left liver lobe) into immunodeficient NSG (NOD SCID gamma, NOD.Cg-*Prkdc^scid^ Il2rg^tm1Wjl^*/SzJ) [37] mice and showed tumor formation with a success rate of 77.8% (P68 2/3, P83 3/3, TFK-1 2/3 mice) and 100% (P68 2/2, P83 2/2 mice), respectively. The subcutaneous xenografts showed progressive tumor growth until euthanasia. Both P68 and P83 present similar tumor volume doubling times of 10.3 ± 3.9 days and 9.9 ± 2.7 days, respectively (Figure 2A,B,D). In contrast to their in vitro behavior, TFK-1 organoid xenografts displayed the slowest tumor growth with a doubling time of 20.0 ± 7.2 days (Figure 2C,D). Upon orthotopic (intrahepatic) injection, 4/4 mice developed intrahepatic CC as demonstrated via ultrasound and necropsy (Figure 2E,F). It was possible to explant growing subcutaneously induced tumors and re-implant them into mice after fragmentation resulting in subcutaneous tumors with highly homogenous size and form (data not shown). These xenografts were used for further in vivo drug testing (see below).

### 2.3. Patient-Derived Organoids Retained the Histological Characteristics of the Parental Tumor

The PDOs and corresponding xenograft tumors retained their primary tumors’ histological features, in particular, strong positivity for CK7 and MUC1 as well as nuclear accumulation of P53 (Figure 3 and Appendix A).

Furthermore, we also examined the CCLs for typical CC markers (Figure 4, Appendix A). All cell lines showed strong expression for CK7 and CK19 and nuclear accumulation of P53 in immunofluorescence studies. Moreover, following their epithelial origin, EpCAM expression was found in all CCLs.

### 2.4. Patient-Derived Organoids Recapitulate Parental Tumor Molecular Features

Protein-coding genes from P68-derived organoids and their parental tumor as well as positive (TFK-1) and negative controls (benign biliary organoids (P119, established from non-cancer bile duct mucosa), full-wall non-cancer bile duct tissue (P113, P121_duct_), and non-cancer bile duct mucosa (P121_mucosa_)) were subjected to RNA sequencing and transcriptomic analysis. After data normalization (Appendix A), a heatmap was created to reflect transcriptomic similarity based on 2770 significantly variable genes (Appendix A and Figure 5A).

The RNA extracted from three wells of organoids was sequenced independently, demonstrating a high degree of correlation between individual culture wells, which was also seen between the samples of other groups (non-cancer tissue, benign biliary organoids, TFK-1 cell line, TFK-1 organoids) (Figure 5A,B). In contrast, there was a considerable transcriptomic distance between the tumor and corresponding organoids, most likely reflecting differences in the microenvironment. Similar effects were seen between non-cancer tissue and benign biliary organoids. The subsequently performed principal component analysis reflected similar results (Figure 5B). The transcriptomes of both benign and malignant organoids were transcriptomically closer than the organoids with their respective tissues of origin. This phenomenon may be a result of the culture conditions and, more importantly, the lack of non-epithelial cells (stromal cells, immune cells, etc.) in organoids, which are inevitably contained in the transcriptomes obtained from biopsies (Figure 5A,B).

To account for the above-described factors causing transcriptomic differences upon the direct comparison of the primary tumor (T) and patient-derived organoids (PDO), differentially expressed genes between tumor and corresponding non-cancer tissue (biliary mucosa) (H) were compared to differentially expressed genes between patient-derived organoids and benign biliary mucosa-derived organoids (BBO). There were 2651 (tumor versus non-cancer mucosa, 548 upregulated, 2103 downregulated), and 1344 (tumor-derived organoids versus benign biliary organoids, 505 upregulated, 839 downregulated) significantly aberrant expressing genes (Figure 5C and Appendix A). Upon functional enrichment analysis, 35 pathways were significantly enriched between malignant and non-malignant cells both in tissue and organoids, including Ascorbate and aldarate metabolism (hsa00053), Bile secretion (hsa04976), MAPK signaling pathway (hsa004010), and PI3K-Akt signaling pathway (hsa04151) (Figure 5D, Appendix A). Besides, 45.3% (359/793) of the enriched gene ontology (GO) terms identified upon comparing CC tissue with non-cancer biliary tissue could be reproduced in the organoid culture (Figure 5E, Appendix A). The respective enriched pathway networks are shown in Figure 5D and Appendix A. Top enriched GO terms are displayed in Figure 5E and Appendix A.

While over-representation analysis shows common differences, it disregards slight but concordant differences between phenotypes. The gene set enrichment analysis (GSEA) directly addresses this limitation. The GSEA was administered to validate further the PDO’s ideal reproduction of the tumor’s molecular characteristics. We first defined the P68 prominent gene set, which consists of 66 genes, and then analyzed it as a background biological term of the following GSEA (Appendix A). Figure 5F,G shows that the P68 derived organoids could significantly recapitulate the parental tumor transcriptomic features, both when we compared PDO with BBO or non-cancer tissue.

Except for RNA sequencing, whole-exome sequencing of the PDOs and the corresponding tumor was applied to explore the genomic similarities. The exome sequencing data revealed a broad overlap between non-synonymous mutations between PDO and primary tumor. In the aggregate, 92.7 % (51/55) of non-synonymous variants from patient 68 primary tumors could be reproduced in the organoids (Appendix A). Other copy number variant plots also show a high degree of similarity between parental tumor and descendent organoids (Appendix A) with no significant genetic drift.

These next-generation sequencing results substantiate that the patient-derived organoids retain transcriptomic features and the mutational landscape of the parental tumor.

### 2.5. Therapeutic Response of the Human Preclinical Cholangiocarcinoma Models

We next investigated the responses of PDOs and corresponding CCLs to anti-cancer agents regularly used in CC (gemcitabine, sorafenib, cisplatin, and doxorubicin) in a PrestoBlue based cell viability assay. While the inhibitory effects of gemcitabine and sorafenib on the CCLs were quite similar, our PDOs presented individual drug responses, with P68 being more resistant (Figure 6A–D). Not surprisingly, monotherapy of cisplatin has a limited effect (Appendix A). Doxorubicin also showed moderate activity. Results are displayed in Appendix A.

In the following in vivo drug response experiment, gemcitabine was used and compared with the control group. After 14 days, treated xenografts showed halted tumor growth while the control group kept gaining size until euthanasia (Figure 6E).

## 3. Discussion

CC represents a rare but highly malignant cancer with subtypes characterized by unique etiology, risk factors, carcinogenesis, and molecular profile [14]. As preclinical models of CC are considerably rare, our biobank and protocol for establishing organoids and cell lines of different CC subtypes are important tools for a better functional characterization of the disease and establishing novel treatments. As potentially curative surgery is an option only for a fraction of patients, novel therapeutic strategies for advanced CC are needed [38]. Furthermore, CC, independent of anatomical subtype, shows a pronounced resistance against standard pharmacological therapies [39]. The preclinical results of sorafenib in vitro and xenograft models [40,41,42,43] could not be confirmed in phase II studies [44,45]. While not part of the first-line chemotherapy regime for unresectable CC, sorafenib elicited responses in small cohorts and cases with tumor progress under gemcitabine treatment, making it a potential second-line treatment option [46,47,48,49]. Doxorubicin is currently used in alternative poly-drug regimens and locoregional chemotherapeutic treatments for advanced CC [50,51]. The pronounced heterogeneity of this disease, combined with a distinct lack of preclinical models including CC cell lines, has hindered the progress in developing precision therapies for years [52]. With the broader availability of next-generation sequencing, many driver mutations and potential drug targets have been characterized in CC [53]. Some of those druggable targets have already led to clinical trials and approved treatments in the clinic, including IDH1 mutations, FGFR2 or NTRK fusions, HER2 amplification, and others [18,38,54]. Our biobank and protocol contribute new preclinical models for functional studies of so-far uncharacterized driver mutations and the analysis of specific resistance mechanisms towards targeted therapies in CC.

Patient-derived cancer organoids (PDOs) present a relatively novel and highly realistic modeling system with various applications in preclinical cancer research, enabling translational individualized treatment regimens [27,55,56,57,58]. Disadvantages of organoid culture include the high economic impact of organoid culture as compared to 2D cell lines (Matrigel, recombinant growth factors, etc.). In addition, as organoid lines generally grow much slower than 2D cell lines (which can, in turn, be interpreted as their more life-like phenotype), their expansion takes more time than when using 2D cell lines. There is only minimal literature available about cholangiocarcinoma PDOs [31,59]. This paper presents two human cholangiocarcinoma organoid lines and their value in preclinical cancer research, one of them (P68) generated from metastatic iCC without common oncogenic driver mutations in *KRAS*, *TP53*, *IDH1*, or *ARID1A*. After optimization of the culture medium, organoid cultures could be maintained for extended periods of time. Moreover, this finding demonstrates that it is possible to generate stable CCLs from 3D cultures and therefore gain more options for high throughput experiments. All cell lines showed diverse morphologic characteristics in 2D and organoid cultures due to the different origins in the bile duct system, indicating divergent underlying carcinogenesis mechanisms. The PDOs showed individual drug responses to established anti-cancer agents, whereas the corresponding CCLs, as less complex in vitro models, responded quite similarly, emphasizing the potential advantage of PDOs over CCLs in therapeutic experiments. Nonetheless, both PDOs and CCLs carry the inherent limitations of cell culture experiments as there are no other cell types such as stromal or immune cells present. The complex interactions between tumor cells, microenvironment, and the host immune system cannot be reproduced in either system [60].

Patient-derived xenografts (PDXs) retain the histopathological and genomic features of primary tumors, especially the tumor heterogeneity, and present valuable models for studying in vivo drug responses and tumor-stroma/immune interactions [23,61]. On the contrary, in addition to ethical considerations, PDXs established with primary patient tissue require substantial resources and time to perform therapeutic experiments compared to PDOs [23,24]. Patient-derived organoid xenografts (PDOXs) show drug responses comparable to the corresponding patient tumors, thus highlighting their value as an alternative in vivo model [62]. We could engraft subcutaneous and orthotopic PDOXs with high success rates. To further validate the applicability of our PDOXs in preclinical drug testing, we treated P68 subcutaneous PDOXs with gemcitabine as the standard drug for advanced CC. Treated mice presented a moderate response to gemcitabine similar to human CC. For experiments requiring a hepatic microenvironment, orthotopic implantation of CC PDOXs may be more suitable than subcutaneous tumors. We demonstrated a high degree of similarity between the primary human tumor and the orthotopic (i.e., intrahepatic) PDOX, making this a suitable model for CC.

Currently, there are limited published sequencing data comparing cholangiocarcinoma PDOs and corresponding tumors [32]. This project conducted RNA sequencing of both sources. We found considerable transcriptomic differences between primary tumors and PDOs. Interestingly, similar effects were found when comparing non-cancer biliary mucosa and biliary mucosa-derived organoids. These differences may have been induced by the effects of culture conditions on the transcriptome. In addition, the presence of immune cells and stromal cells are in the primary tumor tissue, but not the PDOs, may also have influenced the overall transcriptome of primary tumors. Therefore, it is unclear whether the culture conditions lead to altered expression profiles in organoids or contaminating non-epithelial cells altered the expression profiles in tissue biopsies while the “real” tumor transcriptomes are very much seen in the PDO expression datasets. A mixture of both effects is the most likely answer. On the other hand, the overlapping downstream enriched GO terms and pathways between PDOs and primary tumor (i.e., MAPK pathway and PI3K-Akt signaling pathway) in this research demonstrate that the PDOs retained much of the primary tumor transcriptomic functional characteristics. The following GSEA also confirmed that the generated organoids retain patient-specific signatures. The exome sequencing data showed an exceptionally high degree of similarity between PDOs and primary tumors. This is unsurprising, as PDOs represent pure tumor cell cultures, which naturally reflect the genomic landscape of the primary tumor. The few discrepancies may be explained by different tumor regions biopsied for organoid establishment and primary tumor DNA extraction.

To illustrate the consistency of histologic architecture, we examined the IHC profiles of our PDOs and corresponding xenografts compared to their primary tumors. At present, no particular immunohistochemical profile exists for bile duct cancers [63]. Cytokeratins (CK) 7 and 19 can commonly be found in most CCs [64,65]. Some HCCs show focal positivity for either of these cytokeratins but typically do not display mucin production [66]. MUC1 expression in CC is tightly related to dedifferentiation and tumor aggressiveness [67,68]. In this finding, the PDOs and corresponding PDOXs retained the histological characteristics of their parental tumors in H&E stainings with similar positive expression of CK7 and MUC1 as well as nuclear accumulation of P53 in IHC stainings.

The current research established a robust protocol for PDO generation and a biobank from CC samples. We present the molecular characteristics of permanent CC-PDO lines and report a protocol for the transition of PDOs to classical 2D cell lines. A major limitation is the small sample size and the lack of an organoid line of dCC. dCCs are often small tumors as occlusion of the bile duct occurs early during tumor development [69]. The lack of sufficient tissue samples, therefore, prevented the establishment of a dCC organoid line. However, using this protocol, it can be expected that numerous new CC-derived PDOs and CCLs will be reported in the future, thus significantly expanding the toolbox for preclinical research on cholangiocarcinoma.

## 4. Materials and Methods

### 4.1. Human Samples

Cholangiocarcinoma tissue of patients who underwent surgery (January 2015 to May 2018) at the Department of Gastrointestinal, Thoracic, and Vascular Surgery of University Hospital Carl Gustav Carus Dresden, was obtained directly after resection by a board-certified pathologist, stored on ice, and immediately processed as described below. All patients gave their written informed consent prior to inclusion into the study. Detailed information is shown in Table 1. Histopathology work-up of the resected specimens was done routinely at the Department of Pathology of University Hospital Dresden. Only samples with confirmed histopathologic diagnosis of CC were included in this manuscript. In addition, non-cancer bile duct tissue was collected from three other patients (P113, P119, and P121) as the sampling of biliary mucosa for research purposes would have interfered with the pathological assessment of the status of the specimens obtained from patients P68 and P83. For P121, the complete (full-wall) bile duct (P121_duct_) and the isolated mucosa (P121_mucosa_) were used as two individual samples. We established benign bile duct organoids with tissue from P119 using the same protocol for the cholangiocarcinoma samples (more detailed information is given in Appendix A).

### 4.2. Mice

All animal experiments were carried out in strict compliance with German and European Animal Protection Acts and approved by institutional and governmental animal welfare commissions before initiation of the experiments. Mice were housed in a specific pathogen-free environment with a 12 h light/dark cycle and were given ad libitum access to standard laboratory diet and water.

NSG mice (NOD SCID gamma, NOD.Cg-*Prkdc^scid^ Il2rg^tm1Wjl^*/SzJ) [37] were bred in-house.

### 4.3. Organoid Culture

The tumor tissue was minced into small pieces, washed with phosphate-buffered saline (PBS), and digested using Dispase II (Roche) 1 mg/mL and Collagenase type I (Worthington) 0.1 mg/mL at 37 °C for 2 h. During disaggregation, the samples were pipetted up and down every hour. Obtained tissue fragments were washed with PBS, differential centrifugation (200× *g*, 3 min), and then filtered through 100 μm and then 40 μm cell strainers to remove any huge tissue clusters. Next, resuspended the single cells/cell clusters fraction (1 × 10^5^/well) in Matrigel (Corning) and seeded 30 μL per well in a pre-warmed 48-well plate. After solidifying the Matrigel domes (10 min at 37 °C), 200 µL of medium (Appendix A) was added.

Trypsinized TFK-1 cells (1 × 10^5^/well) were resuspended in Matrigel and cultured using identical conditions. To augment the survival time of the organoids, we deviated from the above protocol and modified the medium as described in Appendix A. The medium was changed twice a week. Organoids were passaged every 6–10 days in a 1:2–1:4 split ratio. This study applied the same protocol to generate an organoid line derived from non-cancer bile duct tissue (P119). All cultures were maintained at 37 °C in a 5% CO_2_ atmosphere.

### 4.4. Cell Culture

After organoid lines were established stably (after six passages), organoids were transferred to two-dimensional (2D) culture and maintained as permanent cell lines. 2–4 wells of organoids were dissociated and mixed with cell recovery solution (Corning). The obtained cell pellet was resuspended in an organoid medium (Appendix A) and transferred to a 24-well cell culture plate without Matrigel. After adherence, the medium was replaced with a 1:1 mixture of organoid and 2D medium. During the following passage, cells were subcultured to 6-well, T-25, T-75 subsequently, and were cultured in pure 2D medium (2 parts DMEM + 20% FCS; 1 part K-SFM (all from Life Technologies), supplemented with 100 U/mL penicillin/streptomycin (Thermo Fisher Scientific, Waltham, MA, USA) [36]. TFK-1 was bought from DMSZ (Leibnitz German Collection of Microorganisms and Cell Cultures) and initially cultured as described in the corresponding datasheet using Roswell Park Memorial Institute 1640 medium (RPMI 1640)/10% FCS. For better comparability, the medium of TFK-1 cells was subsequently changed to a 1:1 mixture of 2D medium and RPMI 1640/10% FCS and after the following passage to pure 2D medium. All cell lines were regularly tested for mycoplasma contamination. Examination of the cellular morphology was carried out using an Axio Vert.A1 inverted microscope (Carl Zeiss).

### 4.5. Treatment

Please see Appendix A.

### 4.6. Patient-Derived Organoids Xenografts

For xenografting, 15 complete wells of each organoid line were harvested and resuspended in a mixture of 20 µL Organoid medium with 20 µL Matrigel. Organoids were injected either subcutaneously in both hind legs or orthotopically in the left liver lobe of NSG^TM^ mice. Subcutaneous tumors were measured with calipers, and tumor volume was calculated using the following formula: V = (Width^2^ × Length)/2. The presence of orthotopic tumor growth was confirmed via necropsy and or ultrasound.

### 4.7. Histology and Immunohistochemistry

Organoids were fixed with 4% paraformaldehyde (PFA), embedded in paraffin, and cut into 2.5 µm sections. Hematoxylin and eosin (H&E) staining, as well as immunohistochemistry (IHC) stainings for CK7, TP53, and MUC1, were performed. The 2D cell lines were cultured on chamber slides and fixed with 2% PFA. Immunofluorescence (IF) for CK7, CK19, EpCAM, and TP53 were carried out. All the histology checking procedures followed the standard protocols. The detailed information of antibodies mentioned in this manuscript was listed in Appendix A. Paraffin blocks from primary tumors and xenograft tissue samples were sliced in 2.5 µm sections, and IHC was completed. Imaging was performed with an EVOS FL Auto microscope (Life Technologies) for IHC or TCS SP5 confocal microscope (Leica) for IF studies. More detailed information about sample preparation and staining is given in Appendix A.

### 4.8. Next-Generation Sequencing

This project recruits several groups for transcriptome sequencing, including primary cholangiocarcinoma (P68), corresponding patient-derived organoids (P68), TFK-1 cell lines, TFK-1 organoids, non-cancer bile duct (P113, P121_duct_, P121_mucosa_), and benign biliary organoids (P119). Besides the primary tumor, there are three biological repetitions for each group.

Detailed methodology is described in Appendix A.

### 4.9. Sequencing Data Analysis

Please see Appendix A.

### 4.10. Statistics

All experiments have been performed at least two times independently, with summary data being presented as mean ± standard deviation (SD). Sample size (*n*) values are given in the relevant figures and Appendix A. All statistical analyses were done with GraphPad Prism (Version 9.1.0, GraphPad Software, San Diego, CA, USA) and above version, using Student’s two-tailed unpaired t-test and compared cancer treated versus untreated. We assumed normality and equal distribution of variance between these two groups. Significance was set for *p*-value < 0.050. Doubling time and IC50 calculation were determined with non-linear regression.

## 5. Conclusions

We established human cholangiocarcinoma organoid and corresponding 2D cell lines using a robust protocol for CC organoid generation and culture. The organoids react to treatment and grow as subcutaneous and orthotopic tumors in mice. This methodological study thus reports novel tools to establish organoid lines, 2D cell lines, and mouse models for preclinical cholangiocarcinoma research.

## Figures and Tables

**Figure 1 ijms-22-08675-f001:**
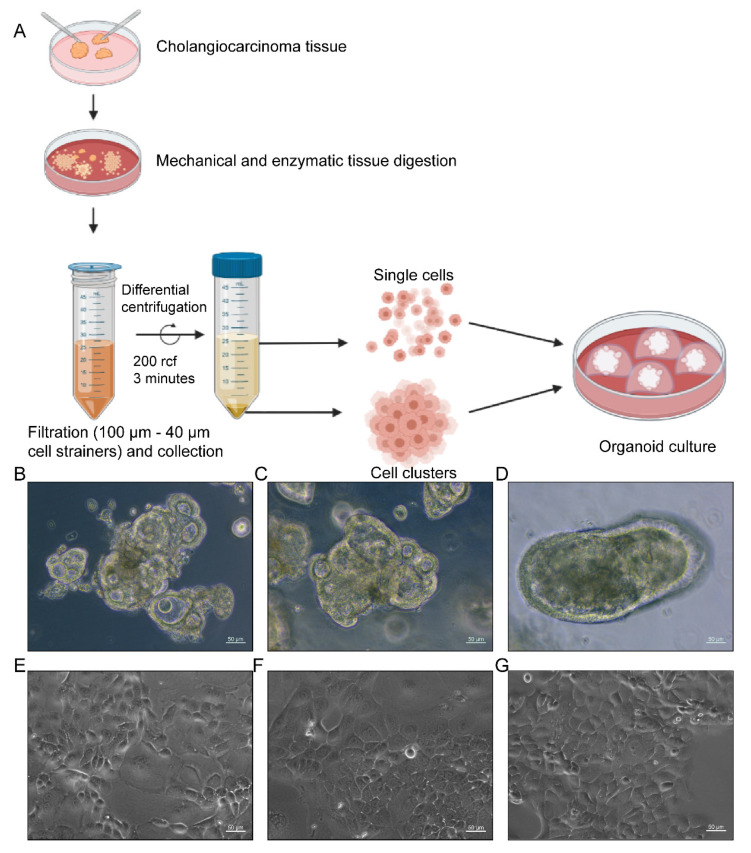
Generation of patient-derived cholangiocarcinoma cell lines. (**A**) Workflow of organoid culture preparation. (**B**–**G**) Top row: Morphology of organoid culture under the phase-contrast microscope (20× magnification): (**B**) P68; (**C**) P83; (**D**) TFK-1; Bottom row: Classical two-dimensional cell culture under the phase-contrast microscope with 20 fold magnification: (**E**) P68; (**F**) P83; (**G**) TFK-1. Scale bar, 50 µm.

**Figure 2 ijms-22-08675-f002:**
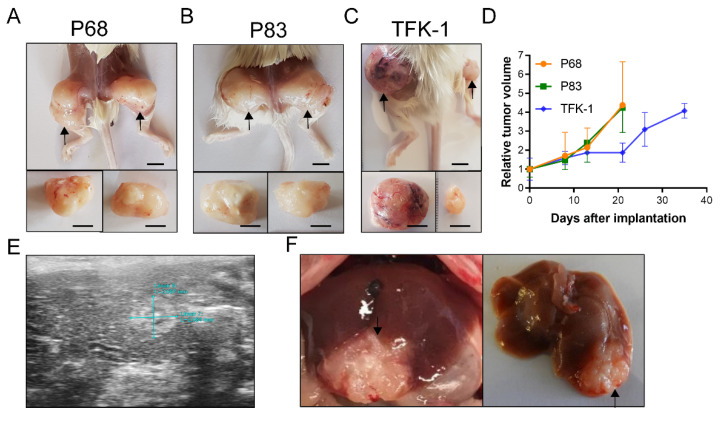
Organoid-derived in vivo xenograft experiments. Top row: Mice with bilateral tumors after subcutaneous injection of PDOs. Pictures from the time point of sacrifice are shown: (**A**) P68; (**B**) P83; (**C**) TFK-1. Scale bar, 5 mm. (**D**) Growth curve of three different organoids in subcutaneous xenografts (*n* = 3). Bottom row: Orthotopic injection of patient-derived organoids in the left liver lobe. (**E**) Ultrasound confirmation of orthotopic tumor formation. (**F**) Orthotopic tumors after dissection.

**Figure 3 ijms-22-08675-f003:**
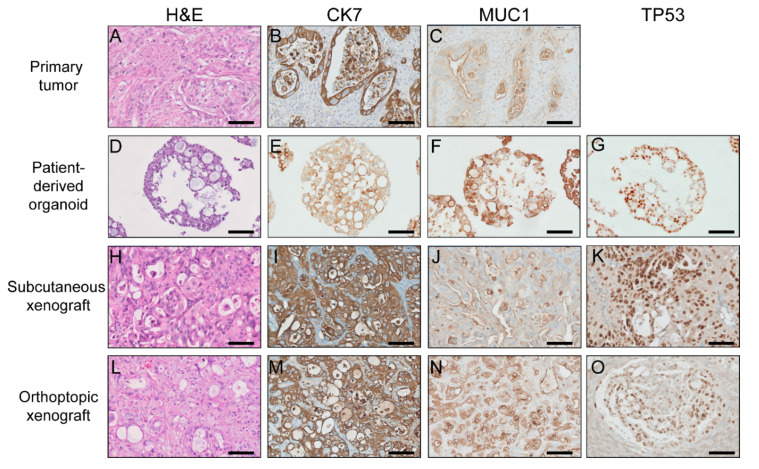
Histologic assessment of parental tumors, patient-derived organoids, and xenograft tumors in patient 83. Hematoxylin and Eosin staining (H&E) (**A**,**D**,**H**,**L**), immunohistochemistry staining of CK7 (**B**,**E**,**I**,**M**), MUC1 (**C**,**F**,**J**,**N**), TP53 (**G**,**K**,**O**) were performed on the primary tumor, patient-derived organoids, subcutaneous xenografts, and orthotopic xenografts. Scale bars, 100 µm.

**Figure 4 ijms-22-08675-f004:**
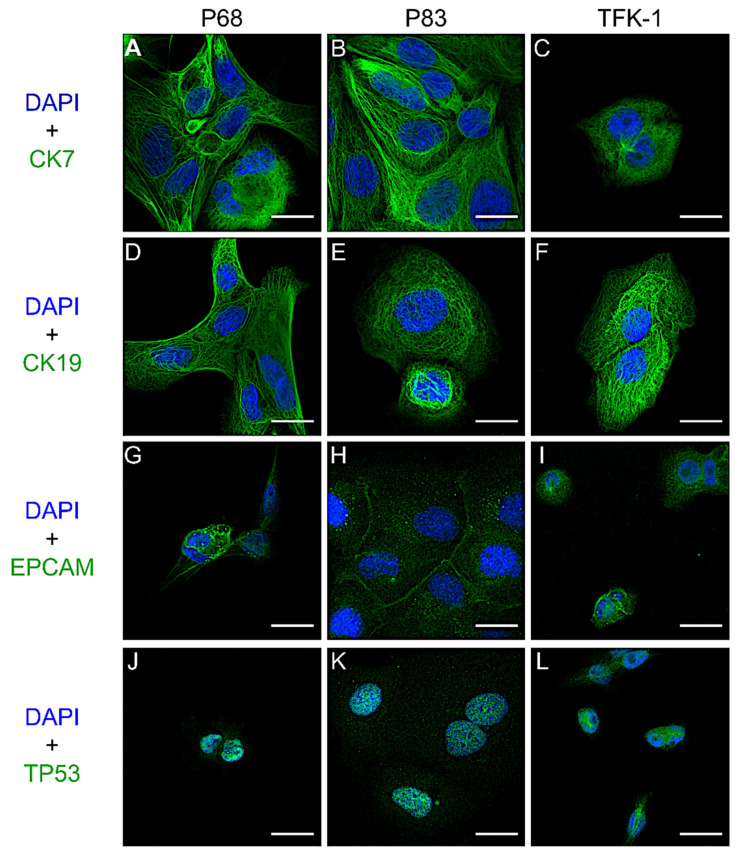
Immunofluorescence staining of patient-derived cell lines. CK7 (**A**–**C**), CK19 (**D**–**F**), EPCAM (**G**–**I**), and TP53 (**J**–**L**) stainings on P68, P83, and TFK-1 cells. Scale bar, 50 µm.

**Figure 5 ijms-22-08675-f005:**
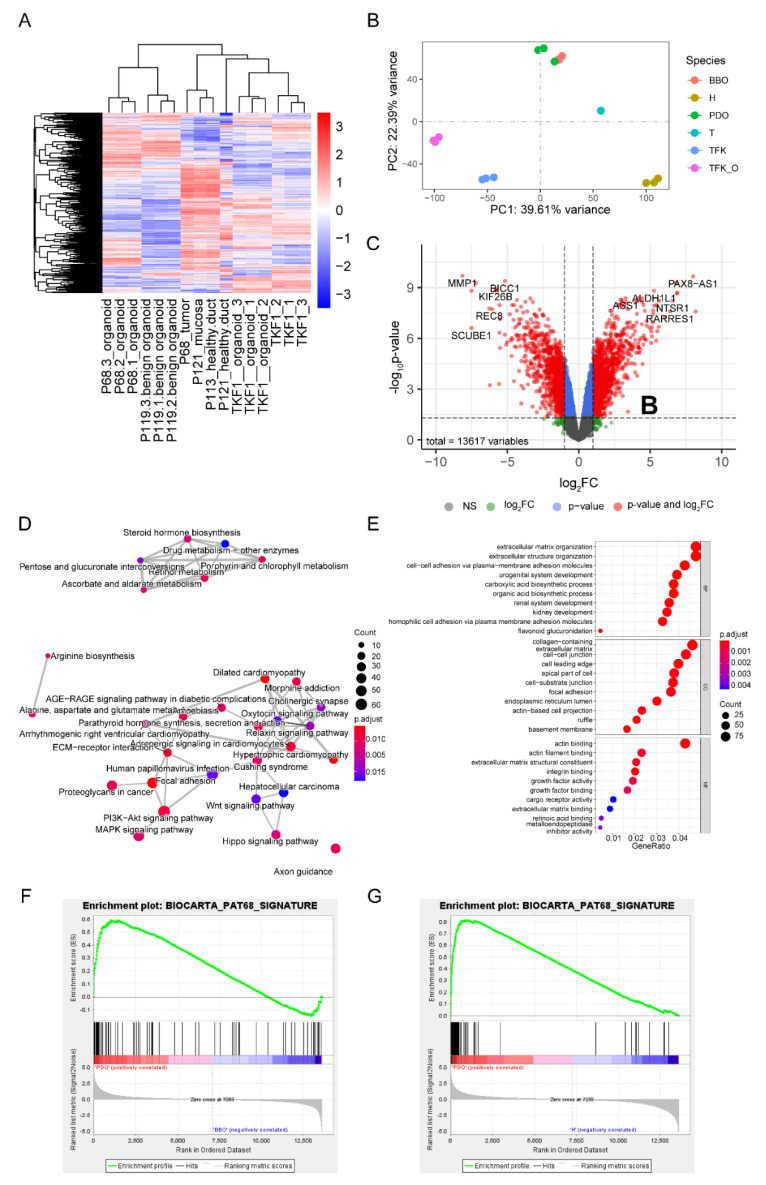
Patient-derived organoids recapitulate features of the primary tumor. (**A**) Heatmap of 2770 significantly regulated genes among all RNA sequencing samples; (**B**) Principal component analysis (PCA) of sequencing samples, each dot represents one sample, x-axis, principal component 1, represents 39.61% variance; y-axis, principal component 2, represents 22.39% variance; (**C**) The volcano plot reflects the existing significantly different expression features between patient-derived organoids and benign biliary organoids. There were 13,617 variable features in total; both *p*-value and fold change significant genes were depicted as red dots while only fold-change positive genes were painted green, genes only with *p* < 0.01 are colored blue, grey represents the remaining non-significant genes. (**D**) Network of significantly enriched pathways in the P68-derived cholangiocarcinoma organoids as compared to benign mucosa organoids. Closely correlated pathways are connected with a line, the size of the dot indicates the number of genes enriched in the respective pathway, the colors represent the adjusted *p*-value. (**E**) Most highly enriched gene ontology terms in P68-derived cholangiocarcinoma organoids. The size of the circular dot reflects the size of each enriched term, and the color indicates the adjusted *p*-values. (**F**,**G**) Gene set enrichment analysis (GSEA) of P68 benign organoids (**F**) and non-cancer tissue (**G**). Abbreviations: BBO, benign biliary organoids; FC, fold change; H, non-cancer tissue; NS, not significant; PC, principal component; PDO, patient-derived organoids; T, tumor; TFK, TFK-1 cell line; TFK_O, TFK-1 organoid culture.

**Figure 6 ijms-22-08675-f006:**
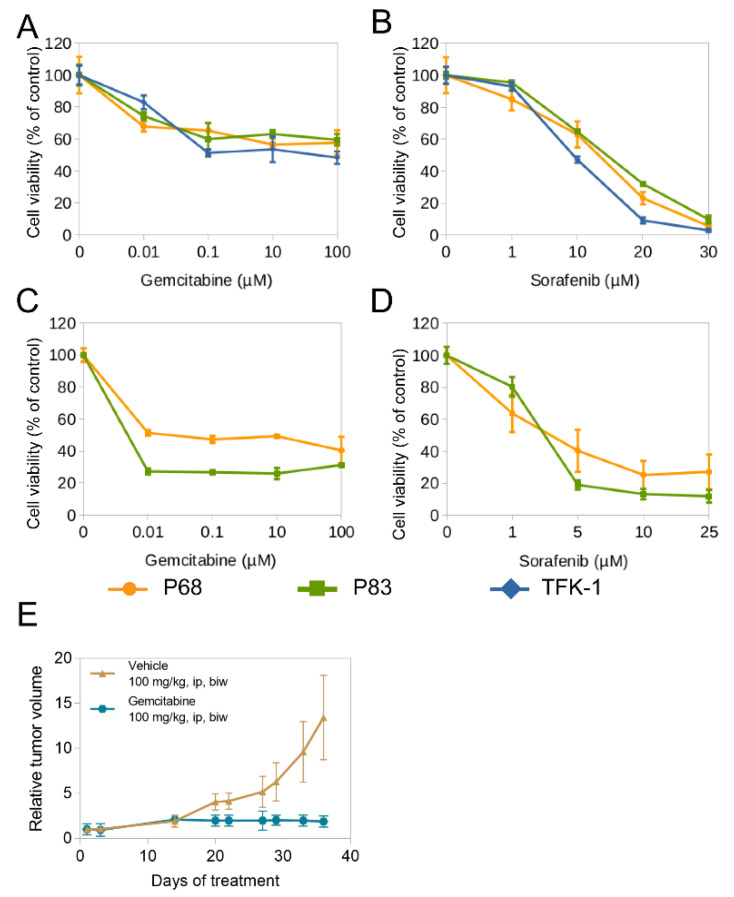
Treatment of PDOs and CCLs. Top row: Dose-response of 2D cell lines treated with (**A**) Gemcitabine; (**B)** Sorafenib; Middle row: Dose-response of organoids with (**C**) Gemcitabine; (**D**) Sorafenib; Bottom: (**E**) Subcutaneous PDO-derived xenografts treated with gemcitabine (100 mg/kg body weight i.p. biw) or glucose solution as control vehicle (*n* = 5 each). Tumor volume was calculated using the following formula: V = (width^2^ × length)/2 and normalized to day 1 of treatment. All experiments were performed in triplicates unless indicated otherwise. Abbreviations: ip, intraperitoneal injection; biw, twice weekly.

**Table 1 ijms-22-08675-t001:** Patient-derived organoid culture records of cholangiocarcinoma.

Patient ID	Age (Years)	Gender	Tumor Type	Isolation Method	Culture Medium *	Culture Period (Weeks)
2	76	M	CC	Manually picked	1	1.1
5	54	M	iCC	Manually picked	1	5.0
5	54	M	iCC	SC	1	5.0
5	54	M	iCC	CCL	1	7.0
7	65	M	iCC	CCL	1	11.3
14	71	F	iCC	CCL	1	7.1
15	56	F	CC	CCL	1	7.1
19	74	M	iCC	CCL	1	7.0
20	76	M	CC	CCL	1	8.7
23	68	M	pCC	CCL	2	8.0
26	26	F	dCC	CCL	2	2.1
42	46	M	CC	CCL + SC	2	6.4
44	56	M	pCC	CCL + SC	2	2.0
50	73	F	CC	CCL + SC	2	6.4
51	72	F	CC	CCL + SC	2	5.9
61	69	F	iCC	CCL	3	35.3
68	57	F	iCC	CCL + SC	3	103.3
70	80	F	iCC	CCL + SC	3	2.0
71	60	M	iCC	CCL + SC	3	3.0
81	82	M	dCC	CCL + SC	3	1.0
83	68	F	pCC	CCL	3	86.4
86	67	F	dCC	SC	3	1.0
95	72	M	iCC	SC	3	2.1
99	63	F	iCC	CCL + SC	3	2.0
108	53	M	iCC	CCL + SC	3	20.1
109	78	F	pCC	CCL + SC	3	15.9
115	65	M	pCC	CCL + SC	3	4.3
118	53	F	iCC	CCL + SC	3	6.6
125	69	M	pCC	CCL + SC	3	7.1

* Culture medium 1, Human stomach organoid culture medium; Culture medium 2, medium 1 + Forskolin + Y-27562; Culture medium 3, medium 2 + Insulin-Transferrin-Selenium; Detailed concentration described in the Appendix A. Abbreviations: M, male; F, female; CC, cholangiocarcinoma; dCC, distal cholangiocarcinoma; iCC, intrahepatic cholangiocarcinoma; pCC, perihilar cholangiocarcinoma; CCL, cell clusters; SC, single cells.

**Table 2 ijms-22-08675-t002:** Clinical information.

Patient-ID	Sex	Age at Diagnosis	Tumor Location	TNM-Stage	Grading
P68	Female	57	Intrahepatic	T2a N1 M1	G2
P83	Female	86	Perihilar	T3 N0 M0	G3
TFK-1	Male	63	Extrahepatic		

## Data Availability

Sequencing data are available upon request.

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
