# Peer review of "Patient-Derived Organoids of Cholangiocarcinoma"

_ijms, 2021, doi:10.3390/ijms22168675_

Round 1

Reviewer 1 Report

Dear Editor, thank you so much for inviting me to revise this manuscript about biliary tract cancer.

The overall limited survival benefit provided by systemic therapies in this setting, with most patients reporting a survival rate of less than a year from the moment of diagnosis, has led to notable efforts towards the identification of novel targets and agents that could modify the natural history of these aggressive hepatobiliary malignancies. In fact, the massive use of next-generation sequencing (NGS) has led to the identification of previously unknown molecular features of CCA, including the presence of specific genetic aberrations that have been suggested to be distinctive features of iCCA and eCCA. Among these druggable alterations, fibroblast growth factor receptor (FGFR)2 gene fusions and rearrangements, isocitrate dehydrogenase-1 (IDH-1) mutations, and BRAF mutations have been widely described in CCA patients, reporting important differences between iCCA and eCCA.

Based on these premises, the paper addresses a timely topic.

The manuscript is quite well written and organized.

Tables are comprehensive and clear.

The introduction explains in a clear and coherent manner the background of this study.

We suggest the following modifications:

  • Although the authors correctly included important papers in this setting, we believe a couple of papers should be cited in the introduction (PMID: 33226854; PMID: 32824407 ; PMID: 33513027; PMID: 33054456), only for a matter of consistency.
  • In addition, we believe some issues deserve further discussion. In everyday clinical practice, we know that the pathologic confirmation of diagnosis is necessary before any non-surgical treatment and can be challenging in BTC, particularly in patients affected by primary sclerosing cholangitis and biliary strictures. In fact, decisions to undertake biopsies should follow a multidisciplinary discussion, especially in potentially resectable tumors. Moreover, endoscopic imaging and tissue sampling are useful but, sadly, biopsy samples are often inadequate for molecular profiling, and in addition, tissue sampling has reported high specificity but low sensitivity in diagnosis of malignant biliary strictures. Finally, the highly desmoplastic nature of BTC limits the accuracy of cytological and pathological approaches. On the basis of these premises, in this scenario, it is urgent to develop new strategies in order to anticipate the diagnosis identifying BTC at an early, resectable stage, and to obtain sufficient material with which to perform genomic analysis. Among these strategies, liquid biopsy has received growing attention over the years, given the promising applications in cancer patients. More specifically, several studies have shown the potential role of liquid biopsy, and the authors should discuss this point, also reporting recent studies in this setting (doi: 10.3390/cells9030721; doi: 10.21873/cgp.20203).

We believe that some revisions are needed. The main strengths of this paper are that it addresses an interesting and very timely question and provides clear answers, with some limitations. We suggest the addition of some references for a matter of consistency. Moreover, the authors should better clarify some points and should add some details and studies, as suggested.

Author Response

Dear Reviewer,

We are grateful for your insightful comments on our manuscript and have revised accordingly. Here is a point-by-point response to your comments and concerns.

  1. “Although the authors correctly included important papers in this setting, we believe a couple of papers should be cited in the introduction (PMID: 33226854; PMID: 32824407 ; PMID: 33513027; PMID: 33054456), only for a matter of consistency.

Response: Thank you for your suggestions. The recommended papers have been included in our manuscript now (References: 18, 28, 30, 33).

  1. “More specifically, several studies have shown the potential role of liquid biopsy, and the authors should discuss this point, also reporting recent studies in this setting (doi: 10.3390/cells9030721; doi: 21873/cgp.20203).”

Response: Thank you very much for this advice. Our group has a strong research interest in liquid biopsy. However, these two papers (Extracellular vesicles, circulating tumor DNA) seem not very relevant to the topic of this manuscript (patient-derived organoids). We would therefore prefer not to cite these papers.

Reviewer 2 Report

The effort made by the authors is important. The text needs improved formatting (lines 124 compared to line 156 or line 156 to 157...) but english is fine. The manuscript shows the effort of a group in getting up and running a pipeline to grow organoids and they show nicely they can grow them and perform different experiments (extract RNA and treat them with different drugs, IHC...). 

The main issue of the manuscript is the novelty and the message. This reviewer is missing more key data to show how useful the model is. As of today there are several published papers with similar systems so there is no novelty regarding this part. In addition the paper should be more succinct showing data and avoiding explaining results not yet shown like in line 218 regarding line 221. Maybe the authors could at least give more data regarding the organoids (genomic alterations and clinical data) so that the organoids can be shared with interested groups. Or if they believe the strength is the protocol itself they can publish as a protocol paper and/or ask for a collaboration with an independent group to confirm the protocol works.

There is no reference in the main text regarding informed consent from patients. This is key for this reviewer.

In addition some points stand:

Fig 1. Not clear what the abbreviations are referring to (lines 117-118).

Line 102 please delete the firs 5 words.

Line 106, there is no reference in the M&M as to when where the patients sample obtained or the project started. Please add some dates. Why did the authors use stomach and not PDAC?

Line 126-7. The authors explain that they had healthy tissue (line 33) and it seems they grew organoids from them. Please clarify this point and explain better.

Line 112 Reading table 1, the culture period of the organoids is not even 12 weeks (3 months) from most of the samples, including medium 3. Please explain better the table and refine the text or delete the word indefinetely. It seems medium 1 works as fine as 3 except for four patients two of which were selected for the experiments. 

Line 124. Only 2 do not even cover the 3 more frequent topological types of CC. Why not choose 3 to increase the relevance as explained in line 82-3? Please amend the text and explain.

Author Response

Dear Reviewer,

We are grateful for your insightful comments on our manuscript and have revised the manuscript accordingly. Here is a point-by-point response to your comments and concerns.

“The text needs improved formatting (lines 124 compared to line 156 or line 156 to 157...)”

Response: Thank you for pointing this out, we apologize for this mistake. It is now corrected.

“In addition the paper should be more succinct showing data and avoiding explaining results not yet shown like in line 218 regarding line 221. Maybe the authors could at least give more data regarding the organoids (genomic alterations and clinical data) so that the organoids can be shared with interested groups. Or if they believe the strength is the protocol itself they can publish as a protocol paper and/or ask for a collaboration with an independent group to confirm the protocol works”

Response: We agree with the reviewer that sufficient data support is necessary. However, we did share all genomic alterations data in the Appendix Tables (Tables A7 and A8), and clinical data in Table 2. The paper is explicitly not meant as a protocol paper.

“There is no reference in the main text regarding informed consent from patients. This is key for this reviewer.”

Response: We agree with the reviewer that informed consent is necessary for this study. Of course, an ethics committee approved the study, and all participants gave their written informed consent prior to enrollment. This was mentioned in line 500 in the original submission. However, we agree with the reviewer about the paramount importance of this and have therefore added another sentence stating that informed consent was given (section 4.1).

“Fig 1. Not clear what the abbreviations are referring to (lines 117-118).”

Response: We apologize for these confusing abbreviations and have corrected the manuscript accordingly.

Line 102 please delete the firs 5 words.”

Response: Thank you for pointing this out. We have corrected it in our revised version of the manuscript.

Line 106, there is no reference in the M&M as to when where the patients sample obtained or the project started. Please add some dates.”

Response: We apologize for the missing dates, we have updated this in our latest version of the manuscript (section 4.1).

“Why did the authors use stomach and not PDAC?

Response: Thank you for this interesting question.

We understand your concern that cells from the bile duct system are more similar to pancreatic cells than gastric tissue. We in fact tried both protocols, and the stomach organoid protocol proved to be superior. We therefore used the stomach protocol rather than the PDAC protocol as a template for the development of the CCC protocol.

The stomach protocol was in development when we started the CCC project, and has been published in the meantime (Seidlitz T, Merker SR, Rothe A, Zakrzewski F, von Neubeck C, Grützmann K, Sommer U, Schweitzer C, Schölch S, Uhlemann H, Gaebler A-M, Werner K, Krause M, Baretton GB, Welsch T, Koo B-K, Aust DE, Klink B, Weitz J, Stange DE. Human gastric cancer modelling using organoids. Gut 2019;68(2):207–17).

“Line 126-7. The authors explain that they had healthy tissue (line 33) and it seems they grew organoids from them. Please clarify this point and explain better.”

Response: We apologize for the confusing phrasing. We generated benign biliary organoids from non-malignant bile duct tissue. The generation and culture settings were the same as for the cancer organoids, but the organoids could not be maintained for longer than a few passages. They were used as benign controls for sequencing procedures. We have modified the manuscript for more clarity in this issue.

“Line 112 Reading table 1, the culture period of the organoids is not even 12 weeks (3 months) from most of the samples, including medium 3. Please explain better the table and refine the text or delete the word indefinetely. It seems medium 1 works as fine as 3 except for four patients two of which were selected for the experiments.”

Response: Thank you for bringing this up. The project aimed to generate PDOs from cholangiocarcinoma. We tried medium 1 initially but failed, so the culture period is already the maximum surviving time of the organoids. Extensive refinement of the media resulted in medium 3. However, for economic reasons, not all organoid lines that grew well in medium 3 were cultured for long periods of time. We therefore chose P68 and P83 as representative organoid lines and froze the others. As a result, most of the culture periods in medium 3 listed in Table 1 are shorter than their maximum theoretical culture. This is the reason why we chose not to calculate and compare average culture times among different media.

This part has been amended accordingly in the manuscript, line 114 -115.

“Line 124. Only 2 do not even cover the 3 more frequent topological types of CC. Why not choose 3 to increase the relevance as explained in line 82-3? Please amend the text and explain.”

Response: Thank you very much for this insightful question.

We agree that to generate one organoid line representing each subtype would have been ideal. However, due to non-scientific disruptive factors, this was not possible. Among those factors was the low case load of distal cholangiocarcinoma in the department; in addition, in the few cases of distal CCC, factors such as small tumor size (forcing the pathologist to claim all tissue for diagnostic procedures rather than research) and thermic damage of the tissue limited the options to generate organoid lines of distal CCC. The manuscript has been amended, and we added this limitation to the text, line 387 - 390.

Reviewer 3 Report

Comment to authors

Overview and General Recommendation:

This manuscript reports a novel approach manuscript reports a protocol of cholangiocarcinoma patient-derived organoid culture as well 32 as a protocol for the transition of 3D organoid lines to 2D cell lines. It is always to see in vitro and in vivo data in the same manuscript, and authors should be applauded for this. Very enjoyable manuscript to read over the weekend. High-quality figures, which make it easier to follow

  1. The introduction summarizes what is available in the literature; please add a few sentences to highlight the novelty of this approach and how it differs from previous reports?
  2. Could the authors comment on the rationale for PDOs were cultured in an organoid medium without Matrigel
  3. "we evaluated the suitability of the PDOs for murine xenograft experiments" in this case, what does stability mean? Please verify for clarity
  4. In the materials section, please list all chemical used and their suppliers. Could you please also provide the methods section solutions, concentrations, instrument settings, and the exact amount added should be stated clearly? Take extra attention to instrumental settings in which the measurement was recorded.
  5. Could the authors include few lines on the Applications of Patient-Derived Organoids
  6. Would there be any limitations to this approach?
  7. In tissue culture experiments, what was the seeding or cellular estimated count?
  8. Organoids derived from non-cancer tissues show more robust proliferation than cancer organoids at the early stage of culture but cease to proliferate. Is it the same case that the authors reported here?
  9. Please ensure space between the numerical value and the units and the same with the last word and the references brackets for clarity
  10. Please leave a space between the numerical value and the units
  11. Some of the relevant references should be included
    1. doi: 3390/cells9040832
    2. DOI:https://doi.org/10.1016/j.celrep.2019.03.088
    3. DOI: 10.1200/JCO.2020.38.4

Author Response

Dear Reviewer,

We are grateful for your insightful comments on our manuscript and have revised the manuscript accordingly. Here is a point-by-point response to your comments and concerns.

  1. “The introduction summarizes what is available in the literature; please add a few sentences to highlight the novelty of this approach and how it differs from previous reports?”

Response: We are grateful for your insightful comment.

There are some studies published in recent years. The total amount of successful cases in long-term culture is still minimal, especially when we specify each subtype, 6 cases of intrahepatic, and only one case of perihilar CCC. We have edited the introduction according to the reviewer’s suggestion, line 92-101.

  1. “Could the authors comment on the rationale for PDOs were cultured in an organoid medium without Matrigel”

Response: Thank you for this interesting question.

To generate 2D cell lines from our human organoids, we resuspended the cell pellet with organoid medium and seeded and switch to the 2D-organoid mixture medium once attached. The purpose of this procedure is to reduce stress on the cells and to gradually adapt them to less rich culture conditions. The detailed method was described in our manuscript, line 434 – 442.

A similar PDO-2D transition protocol was also described by the Tuveson Lab. (http://tuvesonlab.labsites.cshl.edu/wp-content/uploads/sites/49/2018/06/20170523_OrganoidProtocols.pdf)

  1. "we evaluated the suitability of the PDOs for murine xenograft experiments" in this case, what does stability mean? Please verify for clarity

Response: We apologize for the confusing term. We actually evaluated the tumorigenicity of the PDOs. The sentence has been rephrased (line 187 of the updated manuscript).  

  1. “In the materials section, please list all chemical used and their suppliers. Could you please also provide the methods section solutions, concentrations, instrument settings, and the exact amount added should be stated clearly? Take extra attention to instrumental settings in which the measurement was recorded.”

Response: Thank you for this suggestion. We described materials, concentrations in the Appendix 1 file, line 8 - 23. We also add instrumental setting details for the microplate reader, according to your suggestion, Appendix 1, line 27 - 28. 

  1. “Could the authors include few lines on the Applications of Patient-Derived Organoids”

Response: PDO are multifunctional tools for preclinical research. They can be used for the investigation of individual druggable mutations in the context of personalized medicine and for translational studies, e.g., research on early disease detection/prevention. PDOs are also easily applicable for drug testings – with more complex responses than their 2D counterparts. We could show that it is possible to generate PDO-based murine Xenografts and classical 2D cell lines, expanding the range of preclinical CC models. Especially the PDO-based Xenograft model is attractive for in vivo drug response evaluations. We apologize for not having described this in sufficient detail in the manuscript and. described in the manuscript, line 86 – 89.

  1. Would there be any limitations to this approach?

Response: Yes, limitations of organoid culture include the high economic impact of organoid culture as compared to 2D cell lines (Matrigel, recombinant growth factors etc.). In addition, as organoid lines generally grow much slower than 2D cell lines (which can in turn be interpreted as their more life-like phenotype), their expansion takes more time then when using 2D cell lines. This part has been updated in our latest manuscript, line 387 - 393.

  1. In tissue culture experiments, what was the seeding or cellular estimated count?

Response: Thank you for this question.

It is hard to determine the exact cell count as we harvested both cells and clusters after the disintegration of the tumor tissue, but roughly around 100.000 cells per well.

  1. Organoids derived from non-cancer tissues show more robust proliferation than cancer organoids at the early stage of culture but cease to proliferate. Is it the same case that the authors reported here?

Response: We agree with you that it is challenging to generate benign organoid cultures and maintain them long-term.

This study aimed to establish PDOs from cholangiocarcinoma and only developed one benign biliary organoid (P119) and sent them for RNA sequencing as control, without further characterization. Based on this situation, we do not have this information about benign bile duct organoids.

  1. “Please ensure space between the numerical value and the units and the same with the last word and the references brackets for clarity”

Response: We apologize for this omission and have reformatted the manuscript accordingly.

  1. Some of the relevant references should be included
    1. doi: 3390/cells9040832
    2. DOI:https://doi.org/10.1016/j.celrep.2019.03.088
    3. DOI: 10.1200/JCO.2020.38.4

Response: Thank you very much for your suggestions. We have added the first paper (10.3390/cells9040832) to the manuscript (Ref. 31). The second one was already cited in our submission version (10.1016/j.celrep.2019.03.088) (Ref. 59). The last one (10.1200/JCO.2020.38.4) is about hepatocellular carcinoma, which seems not to fit the topic of this manuscript. We would therefore prefer to not cite this article in our manuscript.

Round 2

Reviewer 1 Report

The authors modified the paper according to our suggestions.

We recommend Acceptance in its current form.